# Influence of the Use of Wii Games on Physical Frailty Components in Institutionalized Older Adults

**DOI:** 10.3390/ijerph18052723

**Published:** 2021-03-08

**Authors:** Jerónimo J. González-Bernal, Maha Jahouh, Josefa González-Santos, Juan Mielgo-Ayuso, Diego Fernández-Lázaro, Raúl Soto-Cámara

**Affiliations:** 1Department of Health Sciences, University of Burgos, 09001 Burgos, Spain; jejavier@ubu.es (J.J.G.-B.); rscamara@ubu.es (R.S.-C.); 2Department of Biochemistry, Molecular Biology and Physiology, Faculty of Health Sciences, Campus of Soria, University of Valladolid, 42003 Soria, Spain; diego.fernandez.lazaro@uva.es; 3Neurobiology Research Group, Faculty of Medicine, University of Valladolid, 47005 Valladolid, Spain

**Keywords:** frailty, walking speed, falling risk, balance, Wii, older people, Spain

## Abstract

Aging is a multifactorial physiological phenomenon in which cellular and molecular changes occur. These changes lead to poor locomotion, poor balance, and an increased falling risk. This study aimed to determine the impact and effectiveness of the use of the Wii^®^ game console on improving walking speed and balance, as well as its influence on frailty levels and falling risk, in older adults. A longitudinal study was designed with a pretest/post-test structure. The study population comprised people over 75 years of age who lived in a nursing home or attended a day care center (*n* = 80; 45 women; 84.2 ± 8.7 years). Forty of them were included in the Wii group (20 rehabilitation sessions during 8 consecutive weeks), and the other 40 were in the control group. Falling risk and frailty were evaluated using the Downton scale and Fried scale; balance and walking speed were assessed with the Berg Balance scale and the Gait Speed Test, respectively, as well as the Short Physical Performance Battery (SPPB). The results showed that there was no significant association between Frailty Phenotype and study groups in baseline. However, there was significant association between Frailty Phenotype and study groups at the end of study. Moreover, a significantly higher and negative percentage change (Δ) in the Wii group with respect to the control group on the in falling risk (−20.05 ± 35.14% vs. 7.92 ± 24.53%) and in walking speed (−6.42 ± 8.83% vs. −0.12 ± 4.51%) during study, while there was a higher and positive significant percent change in static balance (6.07 ± 5.74% vs. 2.13 ± 4.64%) and on the SPPB (20.28 ± 20.05% vs. 0.71 ± 7.99%) after 8 weeks of study. The main conclusion of this study was that the use of the Wii^®^ video console for 8 weeks positively influenced walking speed, falling risk, static balance, and frailty levels in older adults. Through a rehabilitation program with the Wii^®^ game console in the older adults, frailty levels are reduced, accompanied by a reduction in falling risk and an increase in static balance and walking speed.

## 1. Introduction

Life expectancy has increased in recent years, which translates into a progressive and remarkable aging of the population. Throughout this process, people undergo a series of cellular and molecular changes [1], which increase the risk of suffering from diseases, a decrease in the quality of life, and the appearance of factors that can cause greater frailty [2]. One of the most relevant aspects to identify alterations in functionality during the aging process is to detect frailty in the older persons. Among these factors are the so-called geriatric syndromes [3], falls being one of the most frequent. This problem is increasing worldwide, and it is estimated that 1 in 3 older adults living in the community suffer one or more falls in a year [4]. In addition, in older people, alterations to multiple physiological systems (dysfunction of the musculoskeletal system, neurological control, and energy metabolism) cause them to present differential physical characteristics, such as poor locomotion or difficulty in complex coordination, all of which are the consequence of poor balance [5]. Impaired postural control is one of the main causes of falls, as this population experiences a decreased ability to maintain and restore balance during physical activities [6]. In this sense, interventions based on physical exercise and balance training strategies are 50% more effective in preventing falls, probably because they promote better static balance, greater mobility, and shorter reaction time [7].

It is noteworthy that, when one of these systems begins to fail or deteriorate, it will not always directly affect balance. However, certain daily activities will be impacted, which present a certain degree of complexity. According to the results of a study carried out in people over 65 years of age, 15% of individuals presented alterations in walking speed, increasing to 35% at 75 years, and it was around 50% in those over 85 years [8]. Impaired walking ability and speed in the older adults can be complicated by falls, which is a predictor of functional impairment, increases morbidity, and contributes to admissions to long-stay nursing homes [9]. For all these reasons, alterations to balance and decompensation in walking speed constitute the most important geriatric syndromes, which lead to falls [3,9]. This close association between poor balance, walking speed, and increased falls suggests the need for activity-based programs that specifically and systematically focus on improving multiple dimensions of the balance system, especially in older adults [10]. In this sense, nonimmersive virtual reality can be used as a promoter of physical activity. Virtual reality promotes interaction between the user and the virtual scene through the performance of body movements captured by specific sensors [11]. The Nintendo Wii^®^ is an example of a video console that uses this type of virtual reality, and this has allowed for the development of new possibilities in systematizing exercise and movement [12].

Several recent investigations have used Wii^®^ games to determine their possible benefits in rehabilitation processes and proprioceptive neuromuscular training, and impact of this type of physical activity on physiological parameters has also been investigated [13,14,15,16]. Along these lines, Nilsagard et al. [17] showed that playing Wii Fit^®^ games twice a week for 6 weeks moderately improved gait and confidence in balance in people with walking problems, such as those with multiple sclerosis. On the other hand, Vieira et al. [16] observed a significant improvement in balance and gait (assessed by the Mini Balance Evaluation Systems Test (Mini-BESTest) and Functional Gait Assessment scales, respectively) in a group of 15 independent frail and prefrail older adults after 14 training sessions, lasting 50 min each, twice a week. On the other hand, Toulotte et al. [18] also found moderate to strong positive effects on static balance, though not dynamic, in a study with 36 noninstitutionalized older adults who played Wii^®^ games for one hour per week for 8 weeks. However, to the authors’ knowledge, whether Wii^®^ games improve frailty physical components in a group such as institutionalized older people has not been sufficiently answered. The older adults group has a special relevance, because up to 50% of them will fall at least once a year as a result of frailty [19], which leads to a decrease in functionality [20]. In this backdrop, taking into account that, in most developed countries, the number of people over 75 years has increased considerably (e.g., Spain [21]) and adverse physiological changes occur during the aging process, which interact with each other and could trigger nonfunctional aging (a consequence of frailty), a precise intervention is necessary to mitigate and/or slow down the levels of frailty in the older adults. Therefore, the objective of this study was to examine the effectiveness of using the Wii^®^ game console for 8 weeks on walking speed and balance, as well as its influence on frailty levels and falling risk, in institutionalized older people.

## 2. Materials and Methods

### 2.1. Participants

A longitudinal study design was employed. The study population comprised 80 people (45 women) over 75 years of age (84.2 ± 8.7 years) who were institutionalized or who attended the “Mixed Nursing Home for the Elderly Burgos I—Cortes” day care center (Burgos, Spain). Participants were selected through nonprobabilistic convenience sampling, accessing the available cases from the population, and a pretest/post-test structure was used. The study was carried out in a natural context, in which attempts were made to control the influence of confounding factors as much as possible.

The inclusion criteria were as follows: being older than 75 years, obtaining a score equal to or greater than 10 in the Lobo Mini-Cognitive Examination (MEC), being institutionalized in the nursing home or going daily to the day center, and being able to stand up with physical support. Subjects who permanently used technical aids (such as wheelchairs), those with a diagnosis of cardiovascular disease, those with hearing and/or visual limitations that prevented them from using the Nintendo Wii^®^ video console, those with severely disorganized behaviors, and those with any type of medical contraindication were excluded.

Participants were randomized to the control group (Control; *n* = 40; 23 women; 83.25 ± 8.78 years) or to the experimental group (Wii; *n* = 40; 22 women; 85.05 ± 8.63 years) by an independent investigator using the block randomization method. In this way, the same number of people in the Wii group as in the control group was ensured. During the course of the research, all participants continued to receive their conventional treatments and therapies provided by the nursing home, such as physical therapy, occupational therapy, and gymnastics sessions. Additionally, the participants of the Wii group received rehabilitation through virtual reality.

### 2.2. Experimental Protocol

Once the participants were selected, different meetings were held with them to present the procedure. In the first meeting, the project, its objective, and the interventions were explained, and participants signed the informed consent form, knowing they were voluntarily accepting to participate in the study. In successive meetings, they were instructed in the world of video games and virtual reality, giving explanations and different demonstrations in order to internalize the use of it.

The intervention consisted of 20 rehabilitation sessions, developed over 8 consecutive weeks, and comprised different activities using the Nintendo Wii Fit^®^ platform. The sessions were distributed so that all participants received two sessions in one week and three sessions in the next, rotating each week cyclically until completing all 20 sessions. Each session lasted 40 min. In the sessions, users worked on different concepts such as balance, gait, stability, aerobic exercises, or stretching of the muscles. In addition, it was intended that users maintained attentional processes in the different activities carried out. At the beginning of the session a therapist explained what the activity consisted of; thereby, as the sessions progressed participants would interact with the game, dispensing with verbal support. Time was allotted at the end of the sessions so participants could discuss their experiences and the impressions they had of the game.

The sessions were held in the assembly hall of the nursing home where there was a television, appropriate material for the Wii^®^ video console, and a chair. Each session followed the same order and was divided into four phases: (I) the two games to be played in that session and their rules were explained to the participants. (II) An aerobic-type game such as “Hula Hoop” was played. In this game, the users were asked to make smooth, wide-radius circles with the hips, simulating the movement of the hula hoop. In addition, the participants were encouraged to raise their arms to shoulder height in order to implicitly increase balance. In this activity, the objectives were to begin the process of interacting with the interface, to focus their attentional processes on the game (since the hula hoop would fall to the ground if they did not pay attention), and to stretch the muscles they would need to use in the next game. (III) A following game was played, specifically to work on balance, such as “Fishing below zero.” In this game, the participants were asked to move from left to right to tilt an ice sheet, and, in this way, the penguin could eat fish. The more fish the penguin ate, the better the score, and better control of static and dynamic balance was achieved. (IV) To end the session, the participants had to choose a game that they wanted to try or play for a period of 5 min. In this last phase, videogames were expected to recover their playful component.

The study was conducted according to the principles of the Declaration of Helsinki and Law 3/2018 of December 5, Protection of Personal Data and Guarantee of Digital Rights, and approved by the Institutional Review Board of the University of Burgos (Protocol code 29/2019, March, 2019).

### 2.3. Main Outcomes—Instruments

All participants underwent an initial assessment at the time of inclusion in the study (T1) and another one 8 weeks later coinciding with the completion of the intervention in the Wii group (T2). This period was decided based on previous research where a Wii console was used by older adults [16,17,18]. The person who made the different assessment visits and the researcher who analyzed the data statistically were blinded with respect to the group to which the participants belonged. In addition, clear instructions were provided to the participants of not revealing the group to which they have been assigned. Their state was evaluated using different instruments and questionnaires.

Static Balance: Static balance was evaluated using the Berg equilibrium scale. This instrument assesses the functional limitations related to the practice of Android Virtual that requires static balance [10]. It is composed of 14 items that represent daily activities the population faces in their lives [22]. Some of them are scored according to the quality of execution, while others are evaluated according to the time required to complete the task [23].The evaluator applied the items and briefly demonstrated each one or read the instructions aloud [22]. During the test, the user’s performance on each task was evaluated with a score between 0 and 4, where 0 was the lowest level of function and 4 the highest, noting additional comments if necessary. Although a score of 0 to 20 was classified as “high falling risk,” “moderate falling risk” from 21 to 41, and “slight falling risk” from 41 to 56; considering this last score as a normal value, this study analyzed static balance by total score. Thus, the higher the score, the better their balance. This is a useful instrument in the older adults and in individuals referred to rehabilitation who present deficits in balance, regardless of age. This tool provides information on the most difficult balancing tasks to perform, facilitates the identification of suitable users for an intervention, and identifies people with the highest falling risk [24]. It is also beneficial for the design of interventions, follow-up, and evaluation of their efficacy. This tool is validated in Spain, with a reliability of 0.98, and is considered one of the best tests for assessing balance at present [23].Walking Speed: To measure walking speed, the “Gait Speed” test was used. To do this, the person walked six or eight meters in a straight line, which required a global walking length of about 10 m, considering the start and end of the walk, with respect to the marked measurement points. The time used to travel that distance “at normal, comfortable speed” was timed and considered the normal value. In primary care, it is usually performed at 3–4 m due to space limitations. It is advised to repeat this process four times and consider the best time [25]. The cutoff points most commonly used to determine the risk threshold are usually between 1 and 0.8 m/sec.Falling Risk: The Downton fall risk was used to assess falling risk [26]. The Downton fall risk index includes 11 risk items, which are scored one point each. Scores were summed to a total index score, ranging 0–11. Although a score of 3 or more indicates a high risk of falling; this study used the total score, where a higher score indicated greater risk of falling (11 was the highest) [23].Physical performance: In the same line, the Short Physical Performance Battery (SPPB or Guralnik test) was used. It consisted of performing three tests: balance (in three positions: feet together, semitandem, and tandem), walk (about 2.4 to 4 m), and get up and sit in a chair five times. It is important to follow the sequence of the tests, since, if the patient starts by standing up and sitting down, he/she can become fatigued and give falsely low performances in the other two subtests. The average administration time, with training, was between 6 and 10 min. The normative values for the Spanish population have been established in various population cohort studies and in primary care. The score and evaluation of the total SPPB result was the sum of the three subtests and ranged from 0 (worst) to 12 (12 indicating the highest degree of lower extremity functioning) [27].Frailty Phenotype: The Frailty Phenotype was determined by Fried Frailty Scale [28]. This scale examines different factors: (a) unintentional weight loss >4.5 kg or >5% in the last year; (b) self-perception of exhaustion; and (c) weakness quantified by means of maximum digital grip strength with a dynamometer adjusted for sex and body mass index (BMI). To assess weakness, the patient was seated, preferably with the dominant hand and the elbow at 90°. The highest value of 3 measurements (separated by one minute) was considered. The cutoff values to determinate weakness for men were BMI ≤ 24 kg/m^2^, grip strength ≤ 29 kg; BMI 24.1–26 kg/m^2^, grip strength ≤ 30 kg; BMI 26.1–28 kg/m^2^, grip strength ≤ 30 kg; and BMI >  28 kg/m^2^, grip strength ≤ 32 kg. For women, the cutoff values were BMI ≤ 23 kg/m^2^, grip strength ≤ 17 kg; BMI 23.1–26 kg/m^2^, grip strength ≤ 17.3 kg; BMI 26.1–29 kg/m^2^, grip strength ≤ 18 kg; and BMI >  29 kg/m^2^, grip strength ≤ 21 kg; (d) Slow gait speed, based on the time required for walking 15 ft (4.57 m) at a normal step. Walk time ≥7 s for men ≤173 cm in height or women ≤159 cm in height or walk time ≥6 s for men whose height was > 173 cm or women whose height was > 159 cm was considered as slowness; (e) Low activity level, energy expenditure of physical activity per week < 383 kcal for men, which corresponds to a minimum of 2.30 h of physical activity per week, or <270 kcal for women, which is equivalent to a minimum of 2 h of activity per week. The presence of 1 or 2 previous characteristics is considered a state of prefrailty, and frailty, by the presence of 3 or more [28,29].

Other data were also collected from participants, such as grip strength, evaluated by dynamometry, or anthropometric parameters, including weight (kg), waist circumference (cm), arm circumference (cm), and leg circumference (cm). Measurements were taken following the standards established by the World Health Organization (WHO).

### 2.4. Statistical Analysis

The data were indicated as mean ± standard deviation of the mean (SD). It calculated the percentage change of each variables between study periods as Δ (%): ((T2 − T1)/T1 × 100). Statistical analysis was completed by SPSS version 25 software (IBM-Inc., Chicago, IL, USA). Moreover, the graph was made by GraphPad Prism 6 software (GraphPad Software, Inc., San Diego, CA, USA). Statistical significance was determined at *p*-value < 0.05.

A two-way repeated-measures analysis of variance (ANOVA) test was performed to explore the interaction effects (time × treatment group: *t* × G) between both groups (control and Wii) along the study for falling risk, frailty, static balance, and walking speed. As age, sex, and abdominal obesity are factors influencing these variables, we included them as possible confounding factors in the different analyses [30,31]. Likewise, the 95% confident interval (95% CI) and statistical power were calculated.

Differences between study periods (from T1 to T2) in each group (control and Wii) were determined by the paired Student´s *t*-test or the Wilcoxon signed rank test, after determinate the normality using the Kolmogorov–Smirnov test. Differences in Δ (%) and other tests at each study period were contrasted between treatment groups by the independent sample Student´s *t*-test or Mann–Whitney U test with treatment group as a fixed factor.

Effect sizes were determined by Ferguson criteria (partial eta squared (η^2^
*p*)), where there is no effect if 0 ≤ η^2^
*p* < 0.05; minimal effect if 0.05 ≤ η^2^
*p* < 0.26; moderate effect if 0.26 ≤ η^2^
*p* < 0.64; and strong effect if η^2^
*p* ≥ 0.64 [32].

Finally, a chi-square test of independence was performed to examine the relation between frailty phenotype and the intervention group in each study period (T1 and T2).

## 3. Results

Table 1 shows the anthropometric data obtained in the sample. When analyzing the relationship between anthropometric values of both groups in the same study period, no statistically significant differences were observed (*p* > 0.05). On the other hand, a statistically significant decrease in waist circumference was obtained in the Wii group after the follow-up time (*p* < 0.05). There were no statistically significant differences between groups over time (all *p* (*t* × G) > 0.05).

A chi-square test of independence showed that there was no significant association between Frailty Phenotype and study group in T1, X^2^ (1, *n* = 80) = 2.739, *p* = 0.098 (Table 2). However, there was significant association between Frailty Phenotype and study group in T2, X^2^ (1, *n* = 80) = 4.267, *p* = 0.039.

Table 3 summarizes the data for static balance, walking speed, and falling risk. When the results obtained in the physical and functional tests in the intragroup ANOVA were compared, significant differences were obtained in the risk of falling and the speed of walking in the intervention group, and in the control group, there was a statistically significant increase in the risk of falling and in equilibrium (*p* < 0.05). On the other hand, no statistically significant differences were observed in the intergroup comparison in the pretest (*p* > 0.05) in any of the variables measured; however, the intervention group had a higher score on the SPPB scale in the final score. When the interrelation between the treatment group and the time (*t* × G) of the physical test scores was obtained, statistically significant differences were obtained in the risk of falling, gait speed, static balance, and SPPB (*p* (*t* × G) < 0.001).

Statistically significant differences were observed in the percentage change of analyzed variables during the study, depending on the group. Specifically, when compared to the control group, the percentage change was higher and negative in the Wii group on falling risk (−20.05 ± 35.14% vs. 7.92 ± 24.53%) and in walking speed (−6.42 ± 8.83% vs. −0.12 ± 4.51%), while it was higher and positive in static balance (6.07 ± 5.74% vs. 2.13 ± 4.64%) and on the SPPB (20.28 ± 20.05% vs. 0.71 ± 7.99%) (Figure 1).

## 4. Discussion

This study proposed to determine the effectiveness of suing Wii Fit^®^ for 8 weeks on walking speed and balance, as well as its influence on frailty level and falling risk, in older people. In addition, we assessed the relationships between frailty and speed, balance, and falling risk. The main result of the study showed that use of Wii Fit^®^ improved walking speed and static balance, and it reduced falling risk and frailty levels.

From medical reports and experience, most people of this age group are frail in their health [5]. Moreover, falls are one of the most disabling geriatric syndromes, which is the result of poor balance and a decrease in walking speed, all of which are a consequence of the appearance of frailty [33]. In this sense, one study analyzed the prevalence of the frailty syndrome and walking speed in the Spanish older population, and frailty was ruled out when the walking speed was greater than 0.9 m/s [34,35]. In this study, 42.6% of the participants had a walking speed lower than 0.8 m/s, 56.4% of which were 75 years or older. This group had a higher risk of frailty (32.1%). On the other hand, in a recent cross-sectional study in 2019, the prevalence of frailty was estimated, and the factors associated with it, such as balance in older patients with type 2 diabetes mellitus, were identified. The prevalence of frailty syndrome was 14.6%, and the monopodal balance test indicated shorter times with the appearance of frailty (*r* = −0.306, *p* < 0.001) [35].

Although structural changes that occur throughout aging are the consequence of physiological factors and multifactorial processes, lifestyle plays a particularly relevant role in determining falling risk and walking speed. Specifically, the WHO advocates nutrition and physical activity as factors that greatly influence frailty levels in older people [36]. Numerous studies have shown that physical activity through specific training or rehabilitation programs, like resistance or suspension training programs, promotes increases in phase angle and handgrip strength in older women [37,38,39]. Moreover, other physical activity programs (translated into the use of the Wii^®^ game console) have presented be capable of reversing, at least partially, physical alterations, such as balance and walking speed in older people [32,33,34].

Nonimmersive virtual reality is being used to promote body movements that are captured by specific physical activity indicators while the user interacts with the virtual scene [11]. Moreover, no studies have reported a negative impact of Wii Fit^®^ training on any measure of balance ability, and most have indicated at least some quantitative or anecdotal evidence of improvement. The therapeutic effect of using the Wii Fit^®^ platform in people over 75 years is on the rise, although the use of virtual reality is not very common in this group, being more frequent in children or adolescents [40]. Therefore, it becomes more difficult to establish comparisons between other variables and samples.

In this line, Cicek et al. [41] compared a physical activity program consisting of a bicycle ergometer and treadmill with an exercise program using the “Nintendo Wii Fit Plus.” The results indicated a significant improvement in the balance test and Timed Up and Go Test (falling risk scale) and improved 10-m walk test in a group of nursing home residents after using the Nintendo Wii Fit Plus^®^ for 30 min twice a week for 8 weeks. Therefore, the authors concluded that both programs demonstrated significant improvement in all parameters, and the video-based program (Wii^®^ game console) was more effective than physical activity, especially in mobility and balance parameters. Likewise, Manlapaz et al. [42] reviewed 16 studies, where the effectiveness of Nintendo Wii Fit™ gaming system protocols for improving balance in healthy older adults (71 and 85 years) was presented. The results obtained from 491 participants (69% females) demonstrated that Wii Fit^®^ exercise games could be a potential alternative to improve balance if technical and safety procedures are provided. At the clinical level, the effective dosage is an important component in any type of intervention, and exergaming should not be an exception. In addition, the authors stated that the exergaming parameters required further research before firm recommendations could be made. Therefore, our results could guide recommendations that an 8-week, controlled Wii^®^ game console intervention could be effective in mitigating the levels of balance and walking speed in the older adults.

On the other hand, to the authors´ knowledge, there is no research that investigates the therapeutic effect of the Wii Fit^®^ video console on frailty and its relationship to falling, walking speed, and SPBB. However, physical performance factors have been strongly associated with decreased frailty, suggesting that improvements in physical performance play an important role in preventing or reducing frailty [43]. Along these lines, a slow walking speed (<0.8 m/s) has been found to be a simple indicator to diagnose frailty in the primary care setting [34]. These data are in line with those displayed in the present study. This could occur, because improving walking speed and balance by using the Wii Fit^®^ would positively affect maintenance of the neuroendocrine and autonomic nervous systems and brain morphology, thus contributing to the prevention of frailty in the older adults [44].

### 4.1. Limitations, Strengths, and Future Lines of Research

The findings of this study must be considered within the context of its limitations. The small sample size and selection using a nonrandomized convenience sampling procedure may lead to the results not being representative of the rest of the population. Likewise, the existence of a small number of studies on this subject makes it difficult to contrast the results obtained. It should be noted that the study presents a series of limitations that would affect the representativeness of the sample and, therefore, the results and conclusions obtained.

It is necessary to propose studies with a larger sample, and at random, to strengthen the hypotheses raised. On the other hand, it is necessary to specify the physiological changes, at the neural level, and of muscular components that are generated as a consequence of the intervention carried out and, if it is possible, to establish comparisons with the physiological changes produced by traditional balance exercises. Therefore, new research is necessary to increase to study and reaffirm the possible relationship between falling risk and balance, and walking speed and balance, among older adults with sarcopenia (according to their age), and to analyze their influence on strength and functional capacity.

### 4.2. Practical Applications

This study presents use of the Nintendo Wii Fit™ game console for 8 weeks (a total of 20 sessions) in older adults as stable therapy in order to improve the physical components related to frailty. In addition, research details the feasibility of other low-cost treatments and opens the possibility of implementing a new innovative rehabilitation method that is accessible, even for services with limited resources. In this way, due to its therapeutic capacity, the Nintendo Wii Fit™ game console is presented as an alternative to conventional therapies, capable of promoting changes in the person, favoring adherence to treatment, and achieving the objectives set within residential centers.

## 5. Conclusions

The main conclusion of this study is that a Wii Fit^®^ console intervention for 8 weeks improved walking speed, static balance, and reduced falling risk and frailty levels in institutionalized older adults. Likewise, current research showed a positive relationship between frailty level changes during the intervention with falling risk and walking speed, and a negative relationship with the Short Physical Performance Battery. Thus, through a rehabilitation program with a Wii Fit^®^ game console, frailty levels were reduced, accompanied by a decrease in falling risk, an increase in static balance, and an increase in walking speed.

## Figures and Tables

**Figure 1 ijerph-18-02723-f001:**
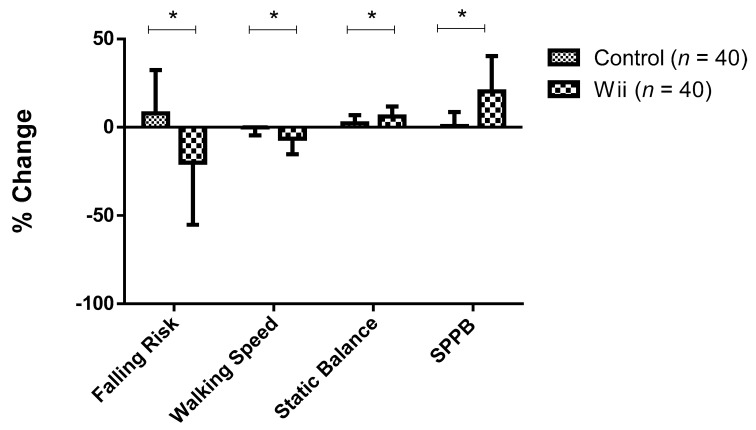
Percentage change for the studied variables in both groups. Data are expressed as mean ± SD. * Significant differences between groups (*p* < 0.001). SPPB = Short Physical Performance Battery.

**Table 1 ijerph-18-02723-t001:** Statistical analysis of the anthropometric parameters.

Study Period	Control (*n* = 40; 23 Women)	95% CI	Wii (*n* = 40; 22 Women)	95% CI	*p*-Value (*t* × G)	η^2^ *p*	Statistical Power
**Body Mass (kg)**							
T1	76.35 ± 13.54	72.15–80.55	74.60 ± 13.01	70.57–78.63	0.366	0.011	0.154
T2	76.18 ± 12.48	72.31–80.05	75.10 ± 12.57	71.20–79.00			
**Waist Circumference (cm)**							
T1	96.73 ± 14.47	92.25–101.21	96.97 ± 14.74	91.40–101.54	0.597	0.004	0.111
T2	96.45 ± 14.16	92.06–100.84	96.63 ± 14.51 *	92.13–101.13			
**Arm Circumference (cm)**							
T1	31.18 ± 37.99	19.41–42.95	31.25 ± 37.99	19.48–43.02	0.172	0.024	0.322
T2	31.31 ± 37.78	16.60–43.02	31.88 ± 38.69	19.89–43.83			
**Leg Circumference (cm)**							
T1	49.86 ± 11.34	46.35–53.37	49.65 ± 12.37	45.82–53.48	0.551	0.005	0.062
T2	49.80 ± 11.13	46.35–53.25	49.51 ± 12.00	45.79–53.23			

Data expressed as mean ± SD; Data adjusted for sex and age. *p*-Value (*t* × G): Group-by-time interaction (*p* < 0.05); Two-factor repeated-measures ANOVA. *: Significant differences between periods within the same study group by dependent Student´s *t*-test (*p* < 0.05); pretest (T1) and post-test (T2).

**Table 2 ijerph-18-02723-t002:** Frailty phenotype distribution in pretest (T1) and post-test (T2) in control and Wii groups.

Frailty Phenotype	Sample	T1		T2	
		Control	Wii	Control	Wii
Prefrailty	*n* (%)	30 (56.6%)	23 (43.4%)	26 (43.3%)	34 (56.7%)
Frailty	*n* (%)	10 (37.0%)	17 (63.0%)	14 (70.0%)	6 (30.0%)
*p*		0.098		0.039	

*p*: Significant differences by chi-square test of independence.

**Table 3 ijerph-18-02723-t003:** Statistical analysis of physical and functional tests.

Study Period	Control (*n* = 40; 23 Women)	95% CI	Wii (*n* = 40; 22 Women)	95% CI	*p*-Value (txG)	η^2^ *p*	Statistical Power
**Right Grip Strength (kg)**							
TI	14.09 ± 6.36	12.12–16.06	13.71 ± 6.18	11.79–15.63	0.650	0.003	0.073
T2	14.54 ± 6.42	12.55–16.53	14.44 ± 6.22	12.51–16.37			
**Left Grip Strength (kg)**							
TI	12.96 ± 5.58	11.23–14.69	12.85 ± 5.80	11.05–14.65	0.196	0.023	0.251
T2	13.29 ± 5.67 *	11.53–15.05	13.52 ± 5.85 *	11.71–15.33			
**Falling Risk (Total Score)**							
TI	3.58 ± 2.09	2.93–4.23	3.75 ± 2.07	3.11–4.39	<0.001	0.248	0.998
T2	3.80 ± 2.17 *	3.13–4.47	2.98 ± 1.76 *	2.43–3.53			
**Walking Speed (sec)**							
TI	20.35 ± 7.40	18.06–22.64	21.24 ± 7.41	18.94–23.54	<0.001	0.198	0.986
T2	20.33 ± 7.45	18.02–22.64	19.96 ± 7.26 *	17.71–22.21			
**Static Balance (Total Score)**							
TI	38.45 ± 11.85	34.78–42.12	39.03 ± 10.67	35.72–42.34	<0.001	0.186	0.979
T2	39.18 ± 12.04 *	35.45–42.91	41.08 ± 10.52 *	37.82–44.34			
**Short Physical Performance Battery (Total Score)**							
TI	6.20 ± 2.23	5.51–6.89	6.33 ± 1.66	5.82–6.84	<0.001	0.208	0.990
T2	6.23 ± 2.24	5.54–6.92	7.70 ± 2.62 *^,&^	6.89–8.51			

Results are showed as mean ± SD. Results were adjusted for age, sex, and waist circumference. *p* (*t* × G): Group-by-Time interaction (*p* < 0.05) calculated by repeated-measures ANOVA. *: Significant differences between study periods in the same group calculated by dependent Student´s *t*-test (*p* < 0.05). ^&^: Significant differences between groups in the same study period calculated by independent Student´s *t*-test (*p* < 0.05).

## Data Availability

Data sharing is not applicable to this article.

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
