# Peer review of "Influence of the Use of Wii Games on Physical Frailty Components in Institutionalized Older Adults"

_ijerph, 2021, doi:10.3390/ijerph18052723_

Round 1

Reviewer 1 Report

After reading the article again, the reviewer agreed to accept the publication of the current article in the IJERPH journal.

Author Response

We would like to sincerely thank the reviewers and editors for their helpful recommendations. We have seriously considered all the comments and carefully revised the manuscript accordingly during the different stages of the review. We feel that the quality of the manuscript has been significantly improved with these modifications and improvements based on the reviewers’ suggestions and comments. We hope our revision will lead to an acceptance of our manuscript for publication in International Journal of Environmental Research and Public Health.

In advance,

Kind regards

Reviewer 2 Report

The authors well addressed all my comments and suggestions. The current version of the manuscript reaches high standards and is suitable for publication.

Author Response

(The authors gave the same response as above.)

Reviewer 3 Report

The authors have adequately addressed all concerns raised and have appropriately revised the initial version of their manuscript.

Author Response

(The authors gave the same response as above.)

Reviewer 4 Report

I have re-reviewed this version of the manuscript and feel that the authors have made all the necessary edits and additions to the content that were requested by reviewers.  I recommend it be considered for publication.

Author Response

(The authors gave the same response as above.)

Reviewer 5 Report

I thank the authors for sharing the results of their research work and congratulations on a well-prepared manuscript.

I have a few minor editorial comments:

Line 219-222 - "2" in the unit for BMI should be superscript;

Table 1, the 95% CI values ​​for Body mass (kg) in the Control group should be without parentheses and the comma should be replaced with dot; please also explain the significance level of the asterisk used for Waist circumference (cm) - it should be done in the description below the table;

Table 2 - please check the writing of all numbers, some have commas instead of dots;

Table 3 - Please explain what significance level is expressed by the asterisk used and &; it should be done in the description below the table;

The description for Figure 2 contains a typo in the word "Percenage" - it should be "Percentage".

Author Response

We would like to sincerely thank the reviewers and editors for their helpful recommendations. We have seriously considered all the comments and carefully revised the manuscript accordingly during the different stages of the review. We feel that the quality of the manuscript has been significantly improved with these modifications and improvements based on the reviewers’ suggestions and comments. We hope our revision will lead to an acceptance of our manuscript for publication in International Journal of Environmental Research and Public Health.

In advance,

Kind regards

This manuscript is a resubmission of an earlier submission. The following is a list of the peer review reports and author responses from that submission.

Round 1

Reviewer 1 Report

I read this article as "Influence of the Use of Wii Games on the Physical Component and Frailty in Institutionalized Older Adults" and found it very interesting. In particular, the impact effects of Wii games is used in the elderly, and this article confirms the impact and effectiveness of the elderly in improving walking speed and balance, as well as its impact on frailty levels and falling risk. These are important topics of concern for the science and technology are used in the field of life. This article submitted on "International Journal of Environmental Research and Public Health (ISSN 1660-4601; ISSN 1661-7827) ". This is a world-class, top journal and is included in the Journal of High Visibility in the Science Citation Index and Social Sciences Citation Index. Therefore, the internal text published is advanced and innovative, as well as a certain degree of contribution. It is therefore suggested that the following should be considered for publication after specific changes have been made. The specific recommendations are as follows:

  1. Please re-check and modify the " p = 443" in line 298.

  1. Please confirm the authenticity of the experimental data again! Based on the 124 lines in the text, the average age of the experimental group (Wii) is 85.05 ± 8.63 years, and the average is very high. From medical reports and experience, most of the old people of this age are frailty in their health. Wii Games has been documented for children in static balance, and the game is designed to be weak in training dynamic balance (walking) and muscle strength. The important conclusion of this article may be the discussion that will be challenged in the future. Although reading this article did not identify deficiencies and avoid future challenges, the author should re-examine the correctness of the data correctly and retained. Finally, the Reviewer recommends that, upon completion of the above comments, agree to accept the publication of the article in International Journal of Environmental Research and Public Health (ISSN 1660-4601; ISSN 1661-7827).

Author Response

Point-by-Point Response to Reviewer’s Comments

We would like to sincerely thank the reviewers for their helpful recommendations. We have seriously considered all the comments and carefully revised the manuscript accordingly. Revisions are highlighted in yellow through the manuscript to indicate where changes have taken place. We feel that the quality of the manuscript has been significantly improved with these modifications and improvements based on the reviewers’ suggestions and comments. We hope our revision will lead to an acceptance of our manuscript for publication in International Journal of Environmental Research and Public Health.

In advance,

Kind regards

Reviewer 1

I read this article as "Influence of the Use of Wii Games on the Physical Component and Frailty in Institutionalized Older Adults" and found it very interesting. In particular, the impact effects of Wii games is used in the elderly, and this article confirms the impact and effectiveness of the elderly in improving walking speed and balance, as well as its impact on frailty levels and falling risk. These are important topics of concern for the science and technology are used in the field of life. This article submitted on "International Journal of Environmental Research and Public Health (ISSN 1660-4601; ISSN 1661-7827) ". This is a world-class, top journal and is included in the Journal of High Visibility in the Science Citation Index and Social Sciences Citation Index. Therefore, the internal text published is advanced and innovative, as well as a certain degree of contribution. It is therefore suggested that the following should be considered for publication after specific changes have been made. The specific recommendations are as follows:

REVIEWER: Please re-check and modify the " p = 443" in line 298.

AUTHORS: Thank you for your observation. The authors have corrected the tip.

REVIEWER: Please confirm the authenticity of the experimental data again! Based on the 124 lines in the text, the average age of the experimental group (Wii) is 85.05 ± 8.63 years, and the average is very high. From medical reports and experience, most of the old people of this age are frailty in their health. Wii Games has been documented for children in static balance, and the game is designed to be weak in training dynamic balance (walking) and muscle strength. The important conclusion of this article may be the discussion that will be challenged in the future. Although reading this article did not identify deficiencies and avoid future challenges, the author should re-examine the correctness of the data correctly and retained. Finally, the Reviewer recommends that, upon completion of the above comments, agree to accept the publication of the article in International Journal of Environmental Research and Public Health (ISSN 1660-4601; ISSN 1661-7827).

AUTHORS: Thank you for your interest. The authors have re-checked the authenticity of the experimental data and these are ok. However, the authors have added a paragraph in the discussion focusing this: “From medical reports and experience, most people of this age group are frail in their health [5]. Moreover, falls are one of the most disabling geriatric syndromes, which is the result of poor balance and a decrease in walking speed, all of which are a consequence of the appearance of frailty [33]. In this sense, one study analyzed the prevalence of the frailty syndrome and walking speed in the Spanish older population, and frailty was ruled out when the walking speed was greater than 0.9 m/s [34,35]. In this study, 42.6% of the participants had a walking speed lower than 0.8 m/s, 56.4% of which were 75 years or older. This group had a higher risk of frailty (32.1%). On the other hand, in a recent cross-sectional study in 2019, the prevalence of frailty was estimated, and the factors associated with it, such as balance in older patients with type 2 diabetes mellitus, were identified. The prevalence of frailty syndrome was 14.6%, and the monopodal balance test indicated shorter times with the appearance of frailty (r = -0.306, p < 0.001) [35].

Reviewer 2 Report

The authors chose an important topic in the aging research area. This topic is too large in the context. There were quite much data presented in the paper, but I do think they should highlight its novelty and analyze their data carefully to reach their research purpose.

While I understand that authors are not native English, a revision of the manuscript is fundamental to improve its readability.

Introduction

The abstract presents acronyms without ever being preceded by full names. Also in this section, no statistical data was reported. Descriptive statistics as well as those relating to the main analyzes should be added.

Introduction

This part is too long to be followed. I have already got lost in the third paragraph. The aim that presented at the end of the introduction part was not clear and needs to be modified.

Methods

My main concern, which leads me to doubt about the suitability of this manuscript for publication, is the methodology.

What is the reason for including people over 75? why were 60/65 to 75 excluded?

Has an a priori analysis been carried out to determine the sample size? Was the interaction of sex on the main variables considered? if not, a 2X2X2 ANOVA would be the most correct approach. Additionally, an ANCOVA should be conducted by entering BMI and age as covariables. Body fat may also affect the results. Similarly, the correlation analysis should be adjusted for the confounding factors mentioned above. Furthermore, the main ANOVA outcomes should be included in the table (confidence intervals and statistical power).

Discussion and conclusion: the authors may reconsider what they wanted to present and modify the discussion part based on the highlights of the results. The conclusion in the current version is not conclusive.

Author Response

Point-by-Point Response to Reviewer’s Comments

We would like to sincerely thank the reviewers for their helpful recommendations. We have seriously considered all the comments and carefully revised the manuscript accordingly. Revisions are highlighted in yellow through the manuscript to indicate where changes have taken place. We feel that the quality of the manuscript has been significantly improved with these modifications and improvements based on the reviewers’ suggestions and comments. We hope our revision will lead to an acceptance of our manuscript for publication in International Journal of Environmental Research and Public Health.

In advance,

Kind regards

Reviewer 2

REVIEWER: The authors chose an important topic in the aging research area. This topic is too large in the context. There were quite much data presented in the paper, but I do think they should highlight its novelty and analyze their data carefully to reach their research purpose.

AUTHORS: Thank you for your recommendation. The authors highlighted the novelty of the study in the introduction section. Likewise, the authors have analyzed newly the data to try reach their research purpose.

REVIEWER: While I understand that authors are not native English, a revision of the manuscript is fundamental to improve its readability.

AUTHORS: Thank you for your recommendation. The manuscript has reviewed by English native.

Abstract

REVIEWER: The abstract presents acronyms without ever being preceded by full names.

AUTHORS: Thank you for your recommendation. The authors have included full names before acronyms.

REVIEWER: Also in this section, no statistical data was reported.

AUTHORS: Thank you for your observation. The authors have included statistical results in the abstract.

REVIEWER: Descriptive statistics as well as those relating to the main analyzes should be added.

AUTHORS: Thank you for your observation. The authors have included descriptive data in the abstract.

Introduction

REVIEWER: This part is too long to be followed. I have already got lost in the third paragraph.

AUTHORS: Thank you for your guidance. Based on your comment and the other reviewers the authors have restructured the introduction in 3 paragraphs.

REVIEWER: The aim that presented at the end of the introduction part was not clear and needs to be modified.

AUTHORS: Thank you for your recommendation. The authors have modified the aim: “Therefore, the objective of this study was to examine the effectiveness of using the Wii® game console for 8 weeks on walking speed and balance, as well as its influence on frailty levels and falling risk, in institutionalized older people. Likewise, the relationships between frailty and speed, balance, and falling risk were investigated.”

Methods

REVIEWER: My main concern, which leads me to doubt about the suitability of this manuscript for publication, is the methodology. What is the reason for including people over 75? why were 60/65 to 75 excluded?

AUTHORS: Thank you for your comment. The authors have included some sentences to explain why we excluded people are less than 75.

“According to the results of a study carried out in people over 65 years of age, 15% of individuals presented alterations in walking speed, increasing to 35% at 75 years, and was around 50% in those over 85 years [8].”

“In this backdrop, taking into account that in most developed countries the number of people over 75 years has increased considerably (e.g., Spain [21]) and adverse physiological changes occur during the aging process, which interact with each other and could trigger non-functional aging (a consequence of frailty), a precise intervention is necessary to mitigate and/or slow down the levels of frailty in the elderly.  “

“In this sense, one study analyzed the prevalence of the frailty syndrome and walking speed in the Spanish older population, and frailty was ruled out when the walking speed was greater than 0.9 m/s [34,35]. In this study, 42.6% of the participants had a walking speed lower than 0.8 m/s, 56.4% of which were 75 years or older.”

REVIEWER: Has an a priori analysis been carried out to determine the sample size?

AUTHORS: Thank you very much for pointing it out. A convenience sample was used to determinate the influence of the Wii uses on walking speed and balance, as well as its influence on the frailty levels and falling risk in in institutionalized older people. This is the reason for the small sample size and the uneven gender distribution. Therefore, the results of this study could be not representatives of the rest of the population. Further, we have added a new paragraph in the limitations section: “The small sample size and selection using a non-randomized convenience sampling procedure may lead to the results not being representative of the rest of the population.”

REVIEWER: Was the interaction of sex on the main variables considered? if not, a 2X2X2 ANOVA would be the most correct approach.

AUTHORS: Thank you for your suggestion. The authors included sex as covariable in the ANCOVA.

REVIEWER: Additionally, an ANCOVA should be conducted by entering BMI and age as covariables. Body fat may also affect the results. Similarly, the correlation analysis should be adjusted for the confounding factors mentioned above.

AUTHORS: Thank you for your recommendation. The authors included in the first-time sex and age like covariables. In the new analysis the authors also have included the waist circumference as indicator of abdominal obesity as a covariables. In this line, Hubbard eta al. (2010) concluded “The association of frailty with a high waist circumference, even among underweight older people, suggests that truncal obesity may be an additional target for intervention.”

The authors have changes the covariables in the statistical section: “As age, sex, and abdominal obesity could be factors influencing these variables, we included them as possible confounding factors in the different analyses [30,31]. Moreover, the 95% confident interval (95% CI) and statistical power were calculated.”

REVIEWER: Furthermore, the main ANOVA outcomes should be included in the table (confidence intervals and statistical power).

AUTHORS: Thank you for your comment. The authors have included confidence intervals and statistical power in the different tables and figures.

REVIEWER: Discussion and conclusion: the authors may reconsider what they wanted to present and modify the discussion part based on the highlights of the results.

AUTHORS: Thank you for your recommendation. the discussion and conclusion have been re-written.

REVIEWER: The conclusion in the current version is not conclusive.

AUTHORS: Thank you for your observation. The authors have modified the conclusion:The main conclusion of this study is that a Wii Fit® console intervention for 8 weeks improved walking speed, balance and gait, and reduced falling risk and frailty levels in institutionalized older people. Likewise, current research showed a positive relationship between frailty level changes during the intervention with falling risk and walking speed, and a negative relationship with the Short Physical Performance Battery. Thus, through a rehabilitation program with a Wii Fit® game console, frailty levels were reduced, accompanied by a decrease in falling risk, an increase in static balance, and an increase in walking speed. Furthermore, there was a relationship between falling risk and walking speed, so it can be said that increasing walking speed decreased falling risk.”

Reviewer 3 Report

Thank you for the opportunity to review this paper. While I believe the topic is an important one. The article is interesting, adequately structured. Nevertheless, before printing, I suggest that the following notes should be considered:

I think the introduction should be improved. The title of this article is about frailty, but the research scope is the falls. So, the authors should clarify the topics and distinguish the difference between those two terms. The authors mentioned “there are few studies that analyze the use of the Wii® in gait and balance in older people” and “there are no references that relate the effect of using the Wii® on frailty in a group such as institutionalized elderly people”. Is it the real situation, or is it because the authors did not give a comprehensive review on “ influence of non-immersive virtual reality game on human health”? Authors should clearly state the research problem or knowledge gap (No previous studies address this issue? Or this issue has not been sufficiently solved? A clear statement of research problem should be provided).

The authors mentioned “all participants underwent an initial assessment at the time of inclusion in the study (T1) and another 8 weeks later (T2), coinciding with the completion of the intervention in the Wii group ”(Lines 160-162). Please explain why you set “8 weeks ”.

Authors may provide more detailed information of sampling, such as why identify the people over 75 years of age as the older adults, while most research used 65 age as threshold according to WHO’s documents.

The English language needs significant improvement. I would suggest getting an English language editor to proofread prior to next round of submission.

Author Response

Point-by-Point Response to Reviewer’s Comments

We would like to sincerely thank the reviewers for their helpful recommendations. We have seriously considered all the comments and carefully revised the manuscript accordingly. Revisions are highlighted in yellow through the manuscript to indicate where changes have taken place. We feel that the quality of the manuscript has been significantly improved with these modifications and improvements based on the reviewers’ suggestions and comments. We hope our revision will lead to an acceptance of our manuscript for publication in International Journal of Environmental Research and Public Health.

In advance,

Kind regards

Reviewer 3

Thank you for the opportunity to review this paper. While I believe the topic is an important one. The article is interesting, adequately structured. Nevertheless, before printing, I suggest that the following notes should be considered:

REVIEWER: I think the introduction should be improved. The title of this article is about frailty, but the research scope is the falls. So, the authors should clarify the topics and distinguish the difference between those two terms.

AUTHORS: Thank you for your observation. To avoid this discrepancy the authors have changed the title by: “Influence of the Use of Wii Games on Physical Frailty Components in Institutionalized Older Adults”

REVIEWER: The authors mentioned “there are few studies that analyze the use of the Wii® in gait and balance in older people” and “there are no references that relate the effect of using the Wii® on frailty in a group such as institutionalized elderly people”. Is it the real situation, or is it because the authors did not give a comprehensive review on “influence of non-immersive virtual reality game on human health”? Authors should clearly state the research problem or knowledge gap (No previous studies address this issue? Or this issue has not been sufficiently solved? A clear statement of research problem should be provided).

AUTHORS: Thank you for your comment. The authors have changed these sentences focusing on that this issue has not been sufficiently solved: “However, to the authors´ knowledge, whether Wii® games improve frailty physical components in a group such as institutionalized elderly people has not been sufficiently answered”

REVIEWER: The authors mentioned “all participants underwent an initial assessment at the time of inclusion in the study (T1) and another 8 weeks later (T2), coinciding with the completion of the intervention in the Wii group”(Lines 160-162). Please explain why you set “8 weeks ”.

AUTHORS: Thank you for your interest. The authors have included this sentence to explain why we set 8 weeks: “This period was decided based on previous research where a Wii console was used by older people [16, 17, 18]. “

REVIEWER: Authors may provide more detailed information of sampling, such as why identify the people over 75 years of age as the older adults, while most research used 65 age as threshold according to WHO’s documents.

AUTHORS: Thank you for your interest. As the reviewer knows, as we get higher the levels of frailty. In this sense, the authors have included several sentences throughout the text indicating the decision to include only people over 75 years of age.

“In this backdrop, taking into account that in most developed countries the number of people over 75 years has increased considerably (e.g., Spain [21]) and adverse physiological changes….”

“In this sense, one study analyzed the prevalence of the frailty syndrome and walking speed in the Spanish older population, and frailty was ruled out when the walking speed was greater than 0.9 m/s [34,35]. In this study, 42.6% of the participants had a walking speed lower than 0.8 m/s, 56.4% of which were 75 years or older.”

REVIEWER: The English language needs significant improvement. I would suggest getting an English language editor to proofread prior to next round of submission.

AUTHORS: Thank you for your recommendation. The manuscript has reviewed by English language editor.

Reviewer 4 Report

The purpose of this study was to determine the efficacy of using a video game console (Wii) in decreasing frailty, along with its relationship with walking speed, balance, and risk of falls. In general, this is a well-designed study and the manuscript is, for the most part, well-written and clearly explained.  It is my recommendation that it be accepted for publication subject to the following relatively minor edits.

  1. Line 40: Change "being the falls" to "falls being".
  2. Lines 59-61: Mismatch of singular and plural:  Delete the 's' from "constitutes" and remove the words, "one of".
  3. Lines 138-139: The phrase, "it was wanted", is quite awkward and confusing, i.e., it was wanted by whom?  Please clarify.
  4. Lines 223-229: This section is difficult to follow and I wonder if it might be better communicated by means of a table?  All the arrows and symbols might confuse readers.
  5. Line 235:  What specifically are the factors to which you are referring?
  6. Line 272: Table 1 - how many males and females are in each group since the number of females is less than 40.
  7. Line 311: Delete "in order" - useless phrase.
  8. LIne 388: Change "New researches are" to "More research is".

Author Response

Point-by-Point Response to Reviewer’s Comments

We would like to sincerely thank the reviewers for their helpful recommendations. We have seriously considered all the comments and carefully revised the manuscript accordingly. Revisions are highlighted in yellow through the manuscript to indicate where changes have taken place. We feel that the quality of the manuscript has been significantly improved with these modifications and improvements based on the reviewers’ suggestions and comments. We hope our revision will lead to an acceptance of our manuscript for publication in International Journal of Environmental Research and Public Health.

In advance,

Kind regards

Reviewer 4

The purpose of this study was to determine the efficacy of using a video game console (Wii) in decreasing frailty, along with its relationship with walking speed, balance, and risk of falls. In general, this is a well-designed study and the manuscript is, for the most part, well-written and clearly explained.  It is my recommendation that it be accepted for publication subject to the following relatively minor edits.

REVIEWER: Line 40: Change "being the falls" to "falls being".

AUTHORS: Thank you for your correction. The authors have resolved this typo.

REVIEWER: Lines 59-61: Mismatch of singular and plural:  Delete the 's' from "constitutes" and remove the words, "one of".

AUTHORS: Thank you for your correction. The authors have resolved this typo.

REVIEWER: Lines 138-139: The phrase, "it was wanted", is quite awkward and confusing, i.e., it was wanted by whom?  Please clarify.

AUTHORS: Thank you for your suggestion. The authors have changed that sentence by: “At the beginning of the session a therapist explained what the activity consisted of; thereby, as the sessions progressed participants would interact with the game, dispensing with verbal support.”

REVIEWER: Lines 223-229: This section is difficult to follow and I wonder if it might be better communicated by means of a table?  All the arrows and symbols might confuse readers.

AUTHORS: Thank you for your recommendation. The authors have modified this section to facilitate the understanding:The cutoff values to determinate weakness for men were BMI ≤24 kg/m2, grip strength ≤29 kg; BMI 24.1–26 kg/m2, grip strength ≤30 kg; BMI 26.1–28 kg/m2, grip strength ≤30 kg; BMI > 28 kg/m2, grip strength ≤32 kg. For women, the cutoff values were BMI ≤23 kg/m2, grip strength ≤17 kg; BMI 23.1–26 kg/m2, grip strength ≤17.3 kg; BMI 26.1–29 kg/m2, grip strength ≤18 kg; BMI > 29 kg/m2, grip strength ≤21 kg; d) Slow gait speed, based on the time required for walking 15 ft (4.57 m) at a normal step. Walk time ≥ 7 s for men ≤173 cm in height or women ≤159 cm in height, or walk time ≥ 6 s for men whose height was > 173 cm or women whose height was > 159 cm, was considered as slowness; e) Low activity level, energy expenditure of physical activity per week < 383 kcal for men, which corresponds to a minimum of 2.30 hours of physical activity per week or < 270 kcal for women, which is equivalent to a minimum of 2 hours of activity per week. Although the presence of 1 or 2 previous characteristics is considered a state of pre-frailty, and frailty by the presence of 3 or more [28,29], the authors considered the total score (5 indicating the highest degree of frailty).

REVIEWER: Line 235:  What specifically are the factors to which you are referring?

AUTHORS: Thank you for your interest. Please see previous comment.

REVIEWER: Line 272: Table 1 - how many males and females are in each group since the number of females is less than 40.

AUTHORS: Thank you for your recommendation. The authors have included the number of women in each study group.

Control (n = 40; 23 women) Wii (n = 40; 22 women)

REVIEWER: Line 311: Delete "in order" - useless phrase.

AUTHORS: Thank you for your recommendation. The authors have deleted “in order”

REVIEWER: LIne 388: Change "New researches are" to "More research is".

AUTHORS: Thank you for your correction. The authors have resolved this typo.

Reviewer 5 Report

Dear authors!

I have read the manuscript titled “Influence of the use of Wii Games on the physical component and frailty in institutionalized older adults” aiming to determine the impact and effectiveness of the use of Wii ® game console in improving frailty as well as its relationship with the walking speed and balance levels and the risk fo falling in institutionalized older people.

Despite finding the manuscript topic interesting, there are several issues to address, such as the language, the organization of the methods, the data analysis (analyzing categorical variables as discrete is a no-no) and interpretation of results (please avoid repeating the data already available in the tables and graphs), the discussion is poorly organized (I suggest guiding your discussion according to Docherty M, Smith R. The case for structuring the discussion of scientific papers. BMJ. 1999;318(7193):1224-5), and the conclusion is lacking clarity.

Author Response

Point-by-Point Response to Reviewer’s Comments

We would like to sincerely thank the reviewers for their helpful recommendations. We have seriously considered all the comments and carefully revised the manuscript accordingly. Revisions are highlighted in yellow through the manuscript to indicate where changes have taken place. We feel that the quality of the manuscript has been significantly improved with these modifications and improvements based on the reviewers’ suggestions and comments. We hope our revision will lead to an acceptance of our manuscript for publication in International Journal of Environmental Research and Public Health.

In advance,

Kind regards

Reviewer 5

I have read the manuscript titled “Influence of the use of Wii Games on the physical component and frailty in institutionalized older adults” aiming to determine the impact and effectiveness of the use of Wii ® game console in improving frailty as well as its relationship with the walking speed and balance levels and the risk fo falling in institutionalized older people.

Despite finding the manuscript topic interesting, there are several issues to address, such as

REVIEWER: the language,

AUTHORS: Thank you for your recommendation. The manuscript has been reviewed by English native.

REVIEWER: the organization of the methods,

AUTHORS: Thank you for your recommendation. The authors haver reorganized the methods section:

2.1. Participants

2.2. Experimental protocol

2.3. Main Outcomes - Instruments

2.4. Statistical analysis

REVIEWER: the data analysis (analyzing categorical variables as discrete is a no-no) and interpretation of results (please avoid repeating the data already available in the tables and graphs),

AUTHORS: Thank you for your recommendation. The authors have used the total scores of different tests in the analysis. In this sense, all analyzed data are discrete avoiding categorical variables. This question is explained in each test in the methods section.

On the other hand, the authors have deleted some data in the result section to avoid repeating the data.

REVIEWER: the discussion is poorly organized (I suggest guiding your discussion according to Docherty M, Smith R. The case for structuring the discussion of scientific papers. BMJ. 1999;318(7193):1224-5), and the conclusion is lacking clarity.

AUTHORS: Thank you for your recommendation. The authors have organized the discussion based on the Docherty et al. (1999) manuscript. In this line, the authors have included limitations, strengths and future research lines, and practical application sections.

4.1. Limitations, Strengths, and Future Lines of Research

The findings of this study must be considered within the context of its limitations. The small sample size and selection using a non-randomized convenience sampling procedure may lead to the results not being representative of the rest of the population. Likewise, the existence of a small number of studies on this subject makes it difficult to contrast the results obtained. These limitations may reduce the representativeness of the findings and may have influenced the results of the study.

Even more emphasis should be placed on implementing more powerful randomized control designs with larger sample populations to test questions of interest. There should also be greater emphasis on determining what specific musculoskeletal and neural adaptations occur in response to Wii Fit® training, and how those changes compare to other commonly used balance training interventions such as uneven balance boards and yoga. Therefore, new research is necessary to increase to study and reaffirm the possible relationship between falling risk and balance, and walking speed and balance, among older people with sarcopenia (according to their age), and to analyze their influence on strength and functional capacity.

4.2. Practical applications

This study presents use of the Nintendo Wii Fit™ game console for 8 weeks (a total of 20 sessions) in older adults as stable therapy in order to improve the physical components related to frailty. In addition, research details the feasibility of other low-cost treatments and opens the possibility of implementing a new innovative rehabilitation method that is accessible, even for services with limited resources. In this way, due to its therapeutic capacity, the Nintendo Wii Fit™ game console is presented as an alternative to conventional therapies, capable of promoting changes in the person, favoring adherence to treatment, and achieving the objectives set within residential centers.

Round 2

Reviewer 2 Report

The authors well addressed my comments. However, the discussion still lacks important comparisons with other training methods proposed for elderly people. An alternative type of exercise is presented in this study. I suggest implementing this section by mentioning other types or training strategies such as those proposed in the following papers:

  • Changes in Phase Angle and Handgrip Strength Induced by Suspension Training in Older Women. Int J Sports Med. 2018 Jun;39(6):442-449. doi: 10.1055/a-0574-3166. 
  • Effects of Pyramid Resistance-Training System with Different Repetition Zones on Cardiovascular Risk Factors in Older Women: A Randomized Controlled Trial. Int J Environ Res Public Health. 2020 Sep; 17(17): 6115.
    Published online 2020 Aug 22. doi: 10.3390/ijerph17176115
  • Effects of Resistance Training with Different Pyramid Systems on Bioimpedance Vector Patterns, Body Composition, and Cellular Health in Older Women: A Randomized Controlled Trial. Sustainability 2020, 12(16), 6658; https://doi.org/10.3390/su12166658

Author Response

Dear Reviewer,

Thank you very much for your suggestions and comments.

We appreciate your time and effort to help us strengthen our manuscript.

Please find below point by point responses to your comments.

In advance,

Kind regards,

REVIEWER 2

REVIEWER: The authors well addressed my comments. However, the discussion still lacks important comparisons with other training methods proposed for elderly people. An alternative type of exercise is presented in this study. I suggest implementing this section by mentioning other types or training strategies such as those proposed in the following papers:

  • Changes in Phase Angle and Handgrip Strength Induced by Suspension Training in Older Women. Int J Sports Med. 2018 Jun;39(6):442-449. doi: 10.1055/a-0574-3166. 
  • Effects of Pyramid Resistance-Training System with Different Repetition Zones on Cardiovascular Risk Factors in Older Women: A Randomized Controlled Trial. Int J Environ Res Public Health. 2020 Sep; 17(17): 6115.
    Published online 2020 Aug 22. doi: 10.3390/ijerph17176115
  • Effects of Resistance Training with Different Pyramid Systems on Bioimpedance Vector Patterns, Body Composition, and Cellular Health in Older Women: A Randomized Controlled Trial. Sustainability 2020, 12(16), 6658; https://doi.org/10.3390/su12166658

AUTHORS: Thank you for your recommendation. The authors have added a sentence to do reference about the resistance and suspension training programs as an alternative of WII: “Numerous studies have shown that physical activity through specific training or rehabilitation programs, like resistance or suspension training programs, promotes increases in phase angle and handgrip strength in older women [37-39].”

Reviewer 5 Report

Dear authors!

I have read the revised version of the manuscript titled “Influence of the use of Wii Games on Physical Frailty Components in institutionalized Older Adults” aiming to examine the effectiveness of using the Wii ® game console for eight weeks on walking speed and balance, as well as its influence on frailty levels and risk of falling, in institutionalized older people; as well as the relationship between frailty and speed, balance, and risk of falling.

I appreciate the modifications after the first review. I do still find a few details which will improve the manuscript. I suggest replacing the expression “Fried Fragility Scale” with “Frailty Phenotype” according to the original publication describing such methods [Fried LP, Tangen CM, Walston J, Newman AB, Hirsch C, Gottdiener J, et al. frailty in older adults: evidence for a phenotype. J Gerontol A Biol Sci Med Sci. 2001;56(3): M146-56. ]. I strongly suggest avoiding the utilization of the expression “elderly” (page 1, line 35, 37, and 45; page 2, line 62, 87, 87, and 94; page 3, line 103; page 4, line 184; page 11, line 361 and 371). Unifying the expression Frailty will improve readability for the manuscript; therefore, I suggest replacing the expression “fragility”. I suggest replacing “non-probability” with “non-probabilistic” (page 3, line 104). I suggest reviewing the language by a professional language editor specialized in health research. Please, double-check page 4, line 188 [ ]. Analyzing categorical variables as continuous variables is absolutely inappropriate.

Author Response

Dear Reviewer,

Thank you very much for your suggestions and comments.

We appreciate your time and effort to help us strengthen our manuscript.

Please find below point by point responses to your comments.

In advance,

Kind regards,

REVIEWER 5

Dear authors!

I have read the revised version of the manuscript titled “Influence of the use of Wii Games on Physical Frailty Components in institutionalized Older Adults” aiming to examine the effectiveness of using the Wii ® game console for eight weeks on walking speed and balance, as well as its influence on frailty levels and risk of falling, in institutionalized older people; as well as the relationship between frailty and speed, balance, and risk of falling.

I appreciate the modifications after the first review. I do still find a few details which will improve the manuscript.

REVIEWER: I suggest replacing the expression “Fried Fragility Scale” with “Frailty Phenotype” according to the original publication describing such methods [Fried LP, Tangen CM, Walston J, Newman AB, Hirsch C, Gottdiener J, et al. frailty in older adults: evidence for a phenotype. J Gerontol A Biol Sci Med Sci. 2001;56(3): M146-56. ].

AUTHORS: Thank you for your suggestion. The authors have made this change along the manuscript.

REVIEWER: I strongly suggest avoiding the utilization of the expression “elderly” (page 1, line 35, 37, and 45; page 2, line 62, 87, 87, and 94; page 3, line 103; page 4, line 184; page 11, line 361 and 371).

AUTHORS: Thank you for your suggestion. The authors have modified the expression “elderly” by “older adults”.

REVIEWER: Unifying the expression Frailty will improve readability for the manuscript; therefore, I suggest replacing the expression “fragility”.

AUTHORS: Thank you for your suggestion. Following your previous suggestion the authors have replaced the expression “fragility” by “frailty”.

REVIEWER: I suggest replacing “non-probability” with “non-probabilistic” (page 3, line 104).

AUTHORS: Thank you for your suggestion. The authors have made this change.

REVIEWER: I suggest reviewing the language by a professional language editor specialized in health research.

AUTHORS: Thanks for your recommendation. The authors sent the manuscript to MDPI English editing services who determined a professional language editor. The authors have attached the MDPI editing service certificate.

Please, double-check page 4, line 188 [ ].

AUTHORS: Thank you for your observation. The authors have included the reference 24 in line 188: “This tool provides information on the most difficult balancing tasks to perform, facilitates the identification of suitable users for an intervention, and identifies people with the highest falling risk [24].”

REVIEWER: Analyzing categorical variables as continuous variables is absolutely inappropriate.

AUTHORS: Thank you very much for your observation. Of course, the categorical variables should not be analyzed as continuous. The article does not analyze categorical variables. The total scores of the different outcomes analyzed are simply collected and analyzed. In this sense, the authors have explained that they used the total score and not categorical variable in all of instruments used.

Although a score of 0 to 20 was classified as "high falling risk", "moderate falling risk" from 21 to 41, and "slight falling risk" from 41 to 56, considering this last score as a normal value, this study analyzed static balance by total score.

The Downton fall risk index includes 11 risk items, which are scored one point each. Scores were summed to a total index score, range 0-11. Although a score of 3 or more indicates a high risk of falling, this study used the total score, where a higher score indicated greater risk of falling (11 was the highest) [23].

Although the presence of 1 or 2 previous characteristics is considered a state of pre-frailty, and frailty by the presence of 3 or more, the authors have considered the total score (ranged 0 to 5; where 5 indicating the highest degree of frailty) [28,29].

Round 3

Reviewer 5 Report

Frailty evaluated with the Frailty Phenotype criteria is not a continuous variable.